# Taylor-Gaussians-Flow: Towards Non-uniform Motion for Novel View Synthesis from Monocular Video

**Zaoming Yan** [1,2]  **Qizhou Chen** [1]  **Yaomin Huang** [1]  **Pengcheng Lei** [1]  **Chenhao Shi** [1]  **Yi Xu** [2]  **Haichuan Song** [1]  **Faming Fang** [1]

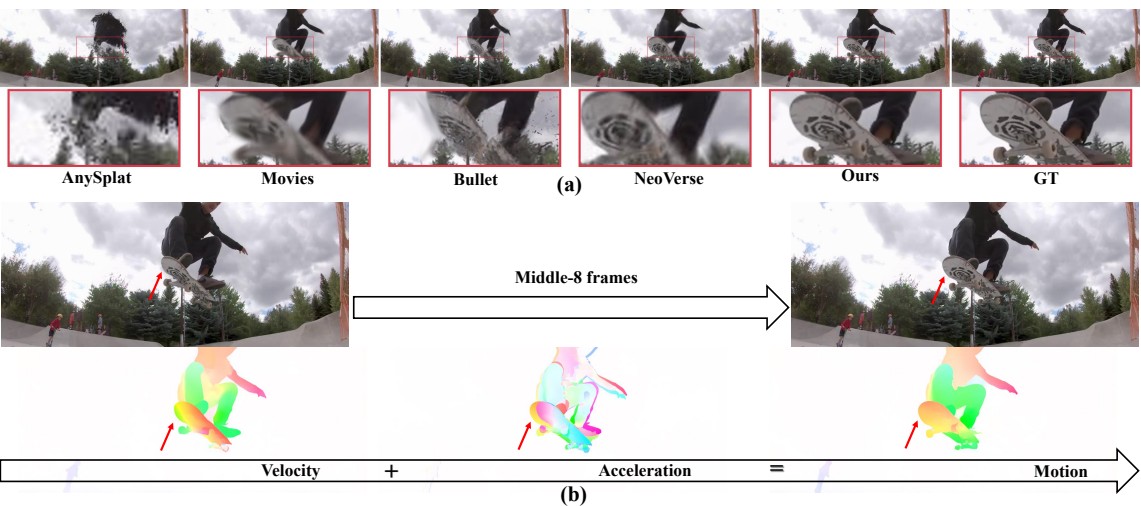

*Figure 1.* **Motivation.** (a) Most Gaussian Splatting methods are built on the first-order approximation (velocity) assumption, but they suffer from non-uniform motion, which becomes more evident in long-term novel view synthesis. (b) Optical flow visualizations demonstrating the necessity of acceleration for modeling long-term non-uniform trajectories. The region indicated by the red arrow demonstrates that acceleration provides complementary motion cues.

## Abstract

Long-term non-uniform motion poses a significant challenge for Novel View Synthesis (**NVS**), as it requires modeling higher-order motion, such as acceleration. Existing methods primarily rely on deformation fields or scene flow, which are limited to first-order approximations. Due to neglecting higher-order motion representations and supervision, these approaches suffer from long-term non-uniform motion scenarios. Inspired by Taylor's theorem, we propose Taylor-Gaussians-Flow (**TGsF**) to represent and supervise non-uniform motion through first-order and

second-order motion components. TGsF comprises two key modules: Taylor-Gaussians (**TGs**) and Taylor-Gaussians-Flow (**TGs-Flow**). TGs represent motion using Gaussian means with a quadratic temporal term and time-dependent opacity. Unlike previous methods, TGs-Flow decouples scene-flow supervision into separate depth and 2D optical-flow constraints. This approach effectively mitigates error propagation from either depth or motion estimation while circumventing the scarcity of labeled scene flow data. Guided by the above analysis, we develop the Feed-Forward Taylor-Gaussians-Flow framework, which sets a new state-of-the-art on four dynamic benchmarks.

## 1. Introduction

Monocular video-based Novel View Synthesis (NVS) enables photorealistic novel view rendering from arbitrary viewpoints and timestamps (Broxton et al., 2020; Collet et al., 2015), supporting key applications such as Virtual Reality (VR) (Gao et al., 2021). Recently, Feed-forward Gaussian Splatting (FF-GS) methods (Charatan et al., 2024;

[1]Department of Computer Science and Technology, East China Normal University, Shanghai, China [2]Midea Group, Shanghai, China. Correspondence to: Zaoming Yan <yan_zaoming@163.com>, Qizhou Chen <52265901009@stu.ecnu.edu.cn>, Haichuan Song <hcsong@cs.ecnu.edu.cn>, Faming Fang <fmfang@cs.ecnu.edu.cn>.

*Proceedings of the $43^{rd}$ International Conference on Machine Learning*, Seoul, South Korea. PMLR 306, 2026. Copyright 2026 by the author(s).

Chen et al., 2024b; Ye et al., 2024; Hong et al.; Jiang et al., 2025) have further advanced NVS by providing high rendering efficiency and quality. However, while these models perform well in regular dynamic scenes, they often fail to accurately capture extremely dynamic scenarios. Specifically, in scenarios such as professional sports events (Smolic et al., 2006), the need for NVS with large temporal displacements highlights the inadequacy of existing motion inference methods (Cho et al., 2025). These scenarios exacerbate the errors in first-order approximation motion estimation inherent in current models.

Dynamic Gaussian Splatting methods can be broadly categorized into two types: deformation-based and scene flow-based approaches. Deformation-based methods (Liang et al., 2025a; Sun et al., 2024; Wu et al., 2025) learn a field to warp Gaussians to target view frames, supervised implicitly by rendering losses. Although these methods enable real-time synthesis (Wu et al., 2025; Lin et al., 2025), they often lack temporal coherence over long-term non-uniform motion sequences. Alternatively, scene flow-based methods (Yang et al., 2026; Lin et al., 2025) explicitly model Gaussian motion to predict future positions. However, these approaches typically rely on a first-order approximation (i.e., constant velocity), leading to significant estimation drift in scenarios with long-term non-uniform motion.

Upon scrutinizing and experimenting on the released implementations of existing methods (Chen et al., 2024b; Lin et al., 2025; Liang et al., 2025a; Yang et al., 2026), we observe significant performance degradation in scenarios with non-uniform motion, as illustrated in Fig. 1(a). We attribute these limitations to the reliance on the first-order approximation for both motion representation and supervision. Specifically: **(i)** Existing methods typically represent motion using only velocity attributes, which lack the higher-order approximation (e.g., acceleration) required to depict non-uniform motion. **(ii)** Current motion loss, primarily based on scene flow loss, is insufficient for non-uniform motion for two reasons. *First*, effective generalization requires large-scale and diverse datasets, yet labeled scene flow data are scarce (Liang et al., 2025b). *Second*, depth and motion are inherently coupled in 2D image-space displacements; consequently, minor inaccuracies in either depth or motion estimation lead to significant errors in 3D scene flow predictions (Liang et al., 2025b).

To address these challenges, we introduce Taylor-Gaussians-Flow (**TGsF**) to mitigate performance degradation in NVS under long-term non-uniform motion. Our core idea is to explicitly represent and supervise motion using a second-order approximation. Specifically, **(i)** to represent non-uniform motion, TGsF incorporates Taylor-Gaussians (**TGs**), which utilize a Taylor-based motion function to model the relationship between time and displacement via both velocity and

acceleration. **(ii)** To circumvent the scarcity of 3D labels and decouple depth from motion, TGsF integrates Flow from Taylor-Gaussians (**TGs-Flow**). Furthermore, TGs-Flow is a differentiable module that projects TGs motion into 2D optical flow. This enables alleviating error propagation in either depth or motion estimation. Error propagation leads to significant errors in 3D scene flow predictions.

Based on these insights, we present the Feed-Forward Taylor-Gaussians-Flow (**FF-TGsF**) framework. The framework consists of two core components: **TGs-Head** and TGs-Flow. TGs-Head, built upon the DPT architecture (Yang et al., 2026), maps features into Taylor-Gaussian parameters. TGs-Flow implements the differentiable projection module described earlier; it analytically computes optical flow from 3D motion, enabling explicit supervision through flow-based losses. We evaluate our method on four dynamic benchmarks: DyCheck-iphone, SNU-HARD, SNU-Extreme, and DAVIS. Extensive ablation studies further quantify the contribution of each component, demonstrating the effectiveness of the proposed FF-TGsF framework in handling non-uniform motion.

The contributions of this work are summarized as: **(a)** We introduce Taylor-Gaussians (TGs) and the TGs-Head to explicitly depict non-uniform motion, providing a second-order approximation of real-world motions. **(b)** We propose a differentiable TGs-Flow module that projects TGs motion into the optical flow domain. This design decouples supervision of depth and motion while circumventing the need for scarce labeled scene flow data. **(c)** Our framework achieves state-of-the-art performance on four challenging dynamic benchmarks: DyCheck-iphone, SNU-HARD, SNU-Extreme, and DAVIS.

## 2. Related Work

### 2.1. Neural Representations For Reconstruction.

Novel view synthesis is an important and challenging task in three-dimensional reconstruction. NeRF (Mildenhall et al., 2021) can effectively learn scene representations and synthesize high-quality novel views through the representation of 3D scenes using neural radiance fields. Numerous subsequent works (Barron et al., 2021)(Barron et al., 2022)(Sun et al., 2022)(Xu et al., 2022) have been proposed to enhance its efficiency and the quality of static scenes. However, challenges persist in the novel view synthesis of dynamic scenes.

To address the challenges of dynamic scenes (Park et al., 2021; Pumarola et al., 2021), the boundaries of novel view synthesis in dynamic settings have been extended. (Fang et al., 2022) proposed modeling temporal information using explicit voxel grids, which accelerates the learning of dynamic scenes to half an hour and is applied in (Guo et al.,

2023; Liu et al., 2023). Optical flow-based methods (Li et al., 2021) (Tian et al., 2023; Zhou et al., 2023; Wu et al., 2024) employ warping algorithms to synthesize novel views by blending nearby frames.

Despite these methods achieving faster training speeds, NeRF-based approaches still face significant challenges in real-time rendering of dynamic scenes, particularly with monocular input.

### 2.2. 3D Gaussian Splatting For Reconstruction

3D Gaussian Splatting (3DGS) (Kerbl et al., 2023) has set a new standard for real-time rendering by representing scenes with anisotropic 3D Gaussians. This paradigm has been adapted to a wide range of tasks, including sparse-view reconstruction (Chen et al., 2024b; Charatan et al., 2024; Xu et al., 2025) and scene editing (Chen et al., 2024a; Zhou et al., 2024).

In dynamic scene reconstruction, a common strategy is to integrate a deformable MLP with 3DGS to model motion (Jung et al., 2023; Guo et al., 2024; Qingming et al., 2025; Lu et al., 2024; Wu et al., 2024; Zhu et al., 2024). Alternatively, some methods treat spacetime as a joint optimization space, representing dynamic scenes with spatiotemporal Gaussian primitives (Yang et al., 2023; Smolak-Dyżewska et al., 2024). However, these spatiotemporal representations often intertwine geometry and motion, complicating the simultaneous optimization of both components.

While significant progress has been made, these approaches typically lack explicit, supervisable motion parameters. Consequently, their underlying motion representations remain insufficient for accurately modeling non-uniform movements.

## 3. Preliminary: 3D Gaussian Splatting

3DGS (Kerbl et al., 2023) associates a 3D Gaussian primitive $i$ with a position $\mu_i$, covariance matrix $\Sigma_i$, opacity $\sigma_i$. The final opacity of a 3D Gaussian at any spatial point $x$ is

$$\alpha_i = \sigma_i \exp\left(-\frac{1}{2}(x - \mu_i)^T \Sigma_i^{-1}(x - \mu_i)\right), \quad (1)$$

where the covariance matrix $\Sigma_i = R_i S_i S_i^T R_i^T$.

To render an image, 3D Gaussians are first projected to 2D image plane via an approximation of the perspective transformation. Specifically, the projection of a 3D Gaussian is approximated as a 2D Gaussian with center $\mu_i^{2D}$ and covariance $\Sigma_i^{2D}$. Center $\mu_i^{2D}$ and covariance $\Sigma_i^{2D}$ are computed as

$$\mu_i^{2D} = \left(K\left((W\mu_i)/(W\mu_i)_z\right)\right)_{1:2}, \quad (2)$$

$$\Sigma_i^{2D} = \left(JW\Sigma_i W^T J^T\right)_{1:2,1:2}, \quad (3)$$

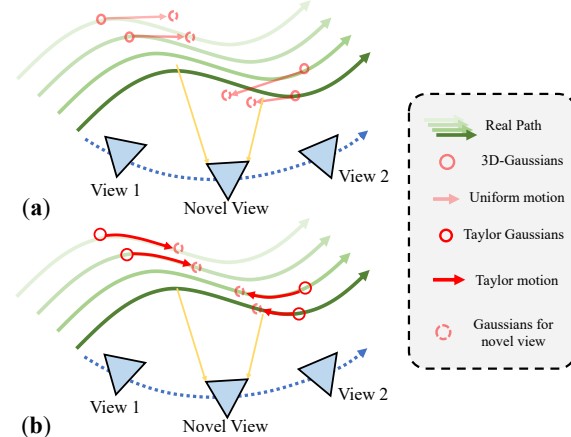

(a)

(b)

*Figure 2.* **Motion Representation.** Comparison between (a) first-order (constant velocity) and (b) second-order (constant acceleration) temporal approximations for modeling Gaussian trajectories.

where $W$, $K$ are the viewing transformation and projection matrix, and $J$ is the Jacobian of the projective transformation.

After sorting the Gaussians in depth order, the color at a pixel is obtained by volume rendering:

$$I(x) = \sum_{i \in \mathcal{N}} c_i \alpha_i^{2D} \prod_{j=1}^{i-1}(1 - \alpha_j^{2D}), \quad (4)$$

where $\mu_i$, $\Sigma_i$, $x$ placed by $\mu_i^{2D}$, $\Sigma_i^{2D}$, $x^{2D}$ (pixel coordinate). $\mathcal{N}$ is the number of Gaussians. Here, $c_i$ is the RGB color after evaluating spherical harmonics (SH) with view direction and coefficients $h_i$. Finally, each Gaussian primitive is optimized by minimizing the photometric loss.

## 4. Method: Taylor-Gaussians-Flow

Existing methods often suffer from inaccurate motion estimation in long-term, non-uniform scenarios, as they rely on first-order approximations (e.g., velocity) and neglect higher-order approximations such as acceleration. To address this, we introduce Taylor-Gaussians-Flow (**TGsF**), which incorporates both first-order and second-order approximations to explicitly represent and supervise non-uniform motion.

In the following sections, we detail: (1) the construction of Taylor-Gaussians (**TGs**) to depict non-uniform motion (Sec. 4.1); and (2) the formulation of differentiable rendering for velocity and acceleration to facilitate explicit supervision (Sec. 4.2).

**Problem Formulation.** For simplicity of notation, we denote the time of the given views as $\{t_k\}_{k=1}^N$, where $N$ is the number of provided views. The goal is to synthesize a novel view at the timestamp $t$. Let $\Delta t = t - t_k$ be the local temporal offset.

## 4.1. Taylor-Gaussians

To represent non-uniform motion, we propose Taylor-Gaussians (**TGs**), as shown in Fig. 2. Specifically, TGs incorporate a Taylor-based motion function into the Gaussian primitives, which establishes the quadratic relationship between time and motion, providing a more accurate approximation of non-uniform motions. Furthermore, we introduce temporal opacity to control the contribution of each TGs primitive over its lifespan. Since the rendering quality of 3D Gaussian Splatting (3D-GS) is more impacted by the opacity and mean of Gaussian primitives (Wang et al., 2025), other parameters of the 3D-GS remain unchanged.

**Taylor motion.** For a 3D spacetime point $(x, t)$, the motion of TGs is defined as:

$$\mu_{i,t_k}(t) = \mu_{i,t_k} + v_{i,t_k}\Delta t + \frac{1}{2}a_{i,t_k}\Delta t^2 \quad (5)$$

where $\mu_{i,t_k}(t)$ denotes the mean of TGs at time $t$. $\mu_{i,t_k}$ is the original mean of the TGs primitive at time $t_k$. $\Delta t$ represents the local temporal offset between the TGs primitive and the view synthesis time. $v_i$ and $a_i$ denote the motion parameters corresponding to the $\Delta t$ and $\Delta t^2$ temporal terms, respectively. $v_i$, $a_i$ are optimized during training. In our implementation, we use the second-order Taylor expansion, as it provides a good balance between representation capacity and rendering time.

**Temporal opacity.** $\sigma_{i,t_k}(t)$ is the temporal opacity that aims to control the impact of the TGs primitive over its lifespan (Chen et al., 2023; Kerbl et al., 2023). We utilize 1D Gaussian for the temporal opacity (Wang et al., 2025) $\alpha_{i,t_k}(t)$:

$$\sigma_{i,t_k}(t) = \sigma_{i,t_k} \cdot \exp\left(-\frac{\Delta t^2}{2s_{i,t_k}}\right), \quad (6)$$

where $s_{i,t_k}$ is temporal scaling factor, and $\sigma_{i,t_k}$ is time-independent TGs opacity at time $t_k$.

## 4.2. Flow from Taylor-Gaussians

Existing methods typically optimize motion trajectories using scene flow losses. However, this approach faces two primary challenges. *First*, depth and motion are inherently coupled in the 2D image plane; since observed 2D displacements are the joint result of both components, minor inaccuracies in one can lead to significant errors in 3D scene flow predictions(Liang et al., 2025b). *Second*, effective generalization requires training on large-scale datasets, yet labeled 3D scene flow data remain scarce. To address these issues, we introduce Taylor-Gaussian-Flow (**TGs-Flow**), a differentiable module that projects 3D Gaussian motion into 2D optical flow. This formulation effectively decouples depth and motion while enabling the use of widely available video data for explicit supervision.

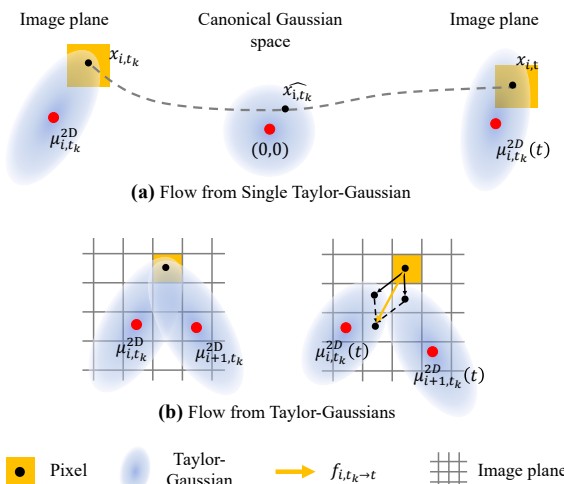

*Figure 3.* **TGs-Flow.** It demonstrates the derivation of optical flow from TGs, which facilitates explicit motion supervision.

**Flow from Single Taylor-Gaussian.** Throughout this section, we denote the reference and target timestamps as $t_k$ and $t$, respectively. Following (Gao et al., 2024), the optical flow $f_{i,t_k \to t}$ is defined as the image-plane displacement of the $i^{th}$ Gaussian primitive between these two time steps:

$$f_{i,t_k \to t} = x_{i,t}^{2D} - x_{t_k}^{2D}, \quad (7)$$

where $x_{t_k}^{2D}$ is a pixel, and $x_{i,t}^{2D}$ is the new projected center following the $i^{th}$ 2D Taylor-Gaussian distribution on the image plane.

We derive the mapping between a single Taylor-Gaussian and its projected pixel displacement as follows. Specifically, we transform the pixel position $x_{t_k}^{2D}$ at reference time $t_k$ into the canonical Gaussian space, and then map it back to the image plane at target time $t$ to determine its new coordinates. This relationship is expressed as:

$$f_{i,t_k \to t} = B_{i,t}B_{i,t_k}^{-1}(x_{t_k}^{2D} - \mu_{i,t_k}^{2D}) + \mu_{i,t_k}^{2D}(t) - x_{t_k}^{2D}, \quad (8)$$

where $\mu_{i,t_k}^{2D}$ and $\mu_{i,t_k}^{2D}(t)$ denote the projected 2D means of the $i^{th}$ Taylor-Gaussian at times $t_k$ and $t$, respectively. The matrix $B_{i,t_k}$ is obtained via the decomposition of the 2D covariance $\Sigma_{i,t_k}^{2D} = B_{i,t_k}B_{i,t_k}^T$.

**Flow from Taylor-Gaussians.** The optical flow at each pixel $x_{t_k}^{2D}$ is computed as the weighted aggregation of individual Gaussian displacements using alpha-composition:

$$\begin{aligned} f_{t_k \to t} &= \sum_{i \in \mathcal{N}} w_i f_{i,t_k \to t} \\ &= \sum_{i \in \mathcal{N}} w_i \left[ B_{i,t}B_{i,t_k}^{-1}(x_{t_k}^{2D} - \mu_{i,t_k}^{2D}) + \mu_{i,t_k}^{2D}(t) - x_{t_k}^{2D} \right], \end{aligned}$$
$$(9)$$

where $\mathcal{N}$ denotes the set of Gaussians intersecting the camera ray, ordered by their depth. The blending weight $w_i$ is

defined as: $w_i = \frac{\alpha_i^{2D} \prod_{j=1}^{i-1}(1-\alpha_j^{2D})}{\sum_i \alpha_i^{2D} \prod_{j=1}^{i-1}(1-\alpha_j^{2D})}$ which ensures the weights are normalized along each ray to facilitate stable supervision.

In long-term non-uniform motion scenarios, dynamic Gaussians may undergo extreme geometric distortion (Ling et al., 2024; Yugay et al., 2023; Matsuki et al., 2024). To mitigate this, we introduce an isotropic regularization term during optimization to keep the scaling factors relatively stable over time. Under the assumption that the Gaussian shape is locally invariant, i.e., $B_{i,t}B_{i,t_k}^{-1} \approx \mathbf{I}$ (where $\mathbf{I} \in \mathbb{R}^{2\times 2}$ is the identity matrix), Eq. (9) simplifies to the difference in projected means:

$$f_{t_k \to t} = \sum_{i \in \mathcal{N}} w_i [\mu_{i,t_k}^{2D}(t) - \mu_{i,t_k}^{2D}] \tag{10}$$

Subsequently, we substitute Eq. (5) into the above equation to obtain the motion offsets for the velocity and acceleration parameters. We employ optical flow to supervise the velocity terms of Eq. (10) and utilize optical flow-derived acceleration to supervise the acceleration terms, as detailed in Sec. 4.4.

### 4.3. Feed-Forward Taylor-Gaussians-Flow

Guided by the above theoretical analysis, we propose FF-TGsF for Novel View Synthesis.

**Backbone.** To ensure stable training, we leverage pretrained 3D foundation models, such as PAGE (Zhou et al., 2025), to provide robust geometric priors for point clouds and camera parameters.

**TGs-Head.** Based on the motion formulation in Sec. 4.1, we design the TGs-Head to decode these features into the parameters of Taylor-Gaussians (TGs). Specifically, while maintaining the standard attributes of 3DGS (Ye et al., 2024), we expand the parameter space to include motion-related parameters (highlighted in red in Fig. 4).

**TGs-Flow.** TGs-Flow serves as a differentiable module that maps these 3D motion parameters into 2D optical flow, following the derivation in Sec. 4.2.

### 4.4. Loss Function

The TGs-Head estimates dynamic TGs parameters from reference frames $\{I_{t_k}\}$ at timestamps $\{t_k\}$ for a given target time $t$. Using the TGs representation, the model renders the target RGB image $I_t$, as well as corresponding depth $D_t$, optical flow $f_{t_k \to t}$, and acceleration $a_{t_k \to t}$. The primary optimization objective is to minimize the reconstruction loss between the rendered image $I_t$ and the ground truth at the target timestamp.

$$\mathcal{L} = \mathcal{L}_{rgb}(\hat{I}, I) + \mathcal{L}_d(\hat{D}, D) + \lambda_f \mathcal{L}_f(\hat{f}, f) + \lambda_a \mathcal{L}_a(\hat{a}, a) + \mathcal{L}_{reg} \tag{11}$$

where $\mathcal{L}_{rgb}$ combines the $L_2$ loss and LPIPS loss. $\mathcal{L}_d$ is a scale and shift-invariant depth loss (Wu et al., 2025). $\mathcal{L}_f$ and $\mathcal{L}_a$ represent the optical flow and acceleration losses, respectively, where the pseudo-ground-truth $\hat{f}$ is generated by a pre-trained model (Morimitsu et al., 2025). $\mathcal{L}_{reg}$ denotes an isotropic regularization term following (Yugay et al., 2023). In motion scenarios, some Gaussians become excessively stretched in scaling (Yugay et al., 2023). To overcome this issue, we introduce an isotropic regularization term during optimization. The loss weights are empirically set to $\lambda_f = 0.7$, $\lambda_a = 0.2$ and $\lambda_{reg} = 0.1$.

To derive the pseudo-ground-truth acceleration $\hat{a_{t_k \to t}}$, we employ a finite difference approximation:

$$\hat{a}_{t_k \to t} = \frac{1}{\Delta t} \left( \frac{f_{t \to t_{k+1}}}{\Delta t} - \frac{f_{t_k \to t}}{\Delta t} \right), \tag{12}$$

where $\hat{f_{t \to t_{k+1}}}$ and $\hat{f_{t_k \to t}}$ denote the estimated optical flows between the respective timestamps.

## 5. Experiments

### 5.1. Experimental Settings

**Fine-tuning dataset.** We follow the fine-tuning strategy (Wu et al., 2025; Zhou et al., 2025) for FF-TGsF on the video datasets DAVIS (Pont-Tuset et al., 2017) and RE10K(Zhou et al., 2018) are treated as pure video datasets without using any pre-calibrated camera information. We utilize pseudo-ground-truth optical flow estimated from the optical flow model (Morimitsu et al., 2025) to supervise the loss function. We follow the official train/validation splits and train only on the training sets. To ensure a fair comparison, we also applied the same fine-tuning strategy to the other models.

**Evaluation Settings.** We follow prior work (Charatan et al., 2024; Jain et al., 2024) and report peak signal-to-noise ratio (PSNR), structural similarity index (SSIM), and LPIPS, all evaluated as in (Charatan et al., 2024; Jain et al., 2024; Wu et al., 2025) for fair comparison. For scene reconstruction, we follow (Charatan et al., 2024; Wu et al., 2025), using two randomly sampled input views with at least 60% overlap and five target views sampled between them.

**Implementation Details.** FF-TGsF is trained on 12 NVIDIA A100 GPUs for approximately 1 day. Input frames are resized to $518 \times 420$ to preserve aspect ratio for pixel-aligned 3DGS prediction. We apply image-level augmentations following EDM (Karras et al., 2022) and optimize using AdamW(Loshchilov & Hutter, 2017) with gradient clipping set to 1.0. We use a batch size of 128, a peak learning rate of $1 \times 10^{-4}$ with 100K linear warm-up iterations, and weight decay of 0.05.

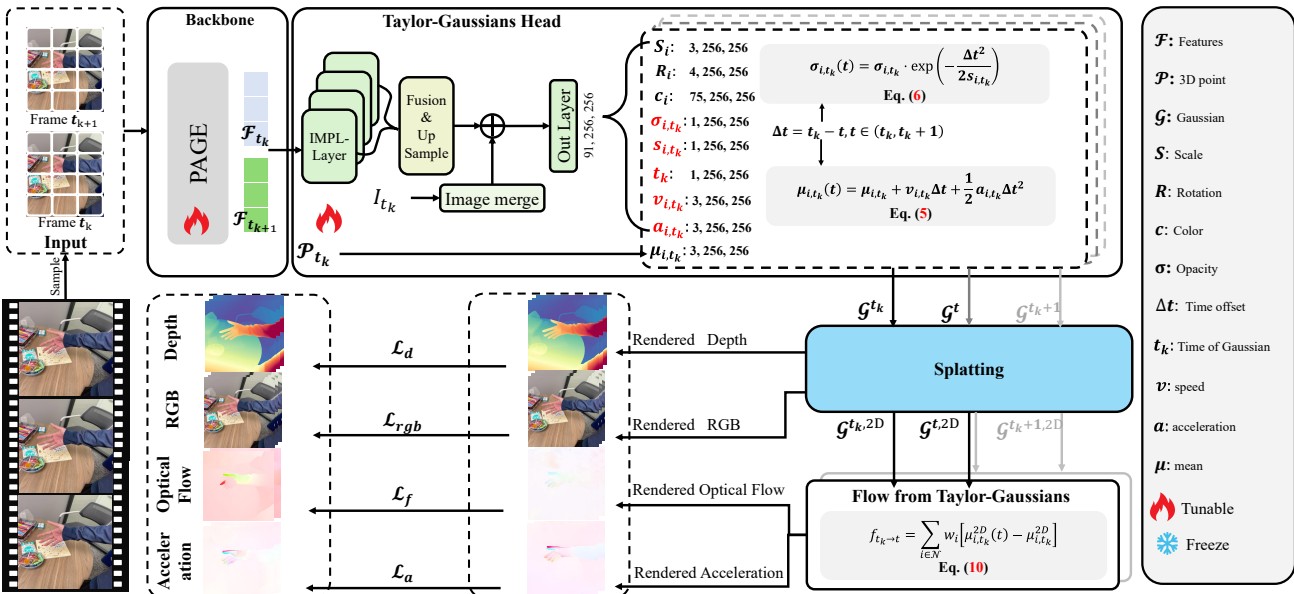

*Figure 4.* **Overall Pipeline.** The Feed-Forward Taylor-Gaussians-Flow framework, integrating the TGs-Head for representation and the TGs-Flow module for supervision.

*Table 1.* Quantitative results are evaluated on DyCheck-iPhone and RE10K datasets. The runtime evaluation is conducted on an A100 GPU. Red shades indicate relative performance gain.

| Method | Met. Type | Scene Rep. | Event year | Rec. Time | DyCheck iPhone | | | RE10K Given View | | | RE10K Novel View | | |
|---|---|---|---|---|---|---|---|---|---|---|---|---|---|
| | | | | | PSNR↑ | SSIM↑ | LPIPS↓ | PSNR↑ | SSIM↑ | LPIPS↓ | PSNR↑ | SSIM↑ | LPIPS↓ |
| pixelSplat(Charatan et al., 2024) | | | CVPR 24 | 0.14 s | 9.53 | 0.219 | 0.905 | 30.70 | 0.952 | 0.055 | 28.31 | 0.905 | 0.097 |
| AnySplat(Jiang et al., 2025) | | | TOG 25 | 0.10 s | 10.16 | 0.233 | 0.881 | 31.48 | 0.962 | 0.046 | 28.48 | 0.909 | 0.091 |
| NoPoSplat(Ye et al., 2024) | Stat. | 3DGS | ICLR 25 | 0.18s | 9.63 | 0.200 | 0.900 | 29.50 | 0.939 | 0.069 | 28.65 | 0.913 | 0.096 |
| PF3plat (Hong et al.) | | | ICML 25 | 0.19s | 7.62 | 0.183 | 0.973 | 23.59 | 0.782 | 0.181 | 23.60 | 0.8494 | 0.182 |
| Splater a Video (Sun et al., 2024) | | | NeurIPS 24 | – | 13.61 | 0.313 | 0.570 | 25.31 | 0.787 | 0.191 | 21.40 | 0.600 | 0.400 |
| StreamSplat(Wu et al., 2025) | | | ICLR 25 | 0.12 s | 18.13 | 0.573 | 0.328 | 31.60 | 0.962 | 0.010 | 24.68 | 0.777 | 0.167 |
| MoVieS (Lin et al., 2025) | Dyn. | 3DGS | CVPR 26 | 0.15 s | 18.46 | 0.589 | 0.309 | 27.37 | 0.810 | 0.089 | 26.66 | 0.802 | 0.088 |
| Bullet (Liang et al., 2025a) | | | NeurIPS 25 | 0.19 s | 16.52 | 0.570 | 0.338 | 26.98 | 0.817 | 0.111 | 25.59 | 0.797 | 0.151 |
| NeoVerse (Yang et al., 2026) | | | CVPR 26 | 0.21 s | 19.56 | 0.593 | 0.258 | 24.11 | 0.730 | 0.050 | 19.40 | 0.560 | 0.066 |
| Ours | | | – | 0.12 s | 20.96 | 0.619 | 0.220 | 28.59 | 0.860 | 0.050 | 26.63 | 0.801 | 0.088 |

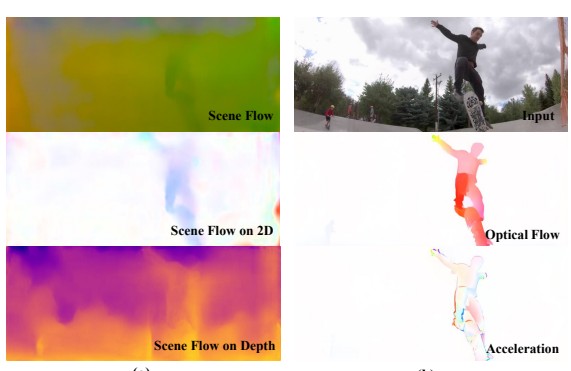

*Figure 5.* Visualization of directly supervised scene flow and decoupled supervised optical flow, under long-term non-uniform motion.

## 5.2. Dynamic novel view synthesis

**Testing datasets.** *DyCheck iPhone*: This benchmark contains 7 dynamic scenes captured by synchronized cameras. Following (Liang et al., 2025a), we use iPhone camera images as context frames and evaluate on the other two stationary cameras. *DAVIS*: A widely adopted benchmark for dynamic NVS consisting of 30 real-world videos. *SNU-HARD/Extreme*: These benchmarks feature significant temporal gaps, making them particularly challenging for long-term synthesis. We evaluate our model on the most difficult configurations.

**Quantitative Results.** Our model consistently outperforms state-of-the-art (SOTA) methods across all metrics on these dynamic benchmarks (Tables 1, 2, and 3). These results demonstrate that our framework provides higher visual fidelity, especially in capturing complex motion details. By incorporating intermediate frames for motion modeling (Wu

*Table 2.* Quantitative results are evaluated on the SNU-HARD and DAVIS benchmarks(Wu et al., 2025).

| Method | SNU-HARD key frames | | | SNU-HARD Middle-8 Frames | | | DAVIS key frames | | | DAVIS Middle-8 Frames | | |
|---|---|---|---|---|---|---|---|---|---|---|---|---|
| | PSNR↑ | SSIM↑ | LPIPS↓ | PSNR↑ | SSIM↑ | LPIPS↓ | PSNR↑ | SSIM↑ | LPIPS↓ | PSNR↑ | SSIM↑ | LPIPS↓ |
| Splater a Video (Sun et al., 2024) | 25.57 | 0.797 | 0.185 | 20.18 | 0.601 | 0.337 | 28.63 | 0.837 | 0.128 | 20.08 | 0.619 | 0.227 |
| StreamSplat (Wu et al., 2025) | 26.18 | 0.814 | 0.168 | 20.15 | 0.598 | 0.312 | 25.83 | 0.712 | 0.196 | 22.10 | 0.613 | 0.234 |
| Movies (Lin et al., 2025) | 26.29 | 0.814 | 0.170 | 21.99 | 0.620 | 0.270 | 27.31 | 0.773 | 0.162 | 22.23 | 0.643 | 0.213 |
| Bullet (Liang et al., 2025a) | 25.58 | 0.790 | 0.124 | 21.06 | 0.584 | 0.291 | 27.20 | 0.790 | 0.157 | 19.98 | 0.572 | 0.276 |
| NeoVerse (Yang et al., 2026) | 26.35 | 0.805 | 0.121 | 22.64 | 0.661 | 0.257 | 29.37 | 0.891 | 0.050 | 23.26 | 0.702 | 0.210 |
| Ours | 28.73 | 0.861 | 0.106 | 25.19 | 0.753 | 0.150 | 30.11 | 0.900 | 0.071 | 24.92 | 0.743 | 0.162 |

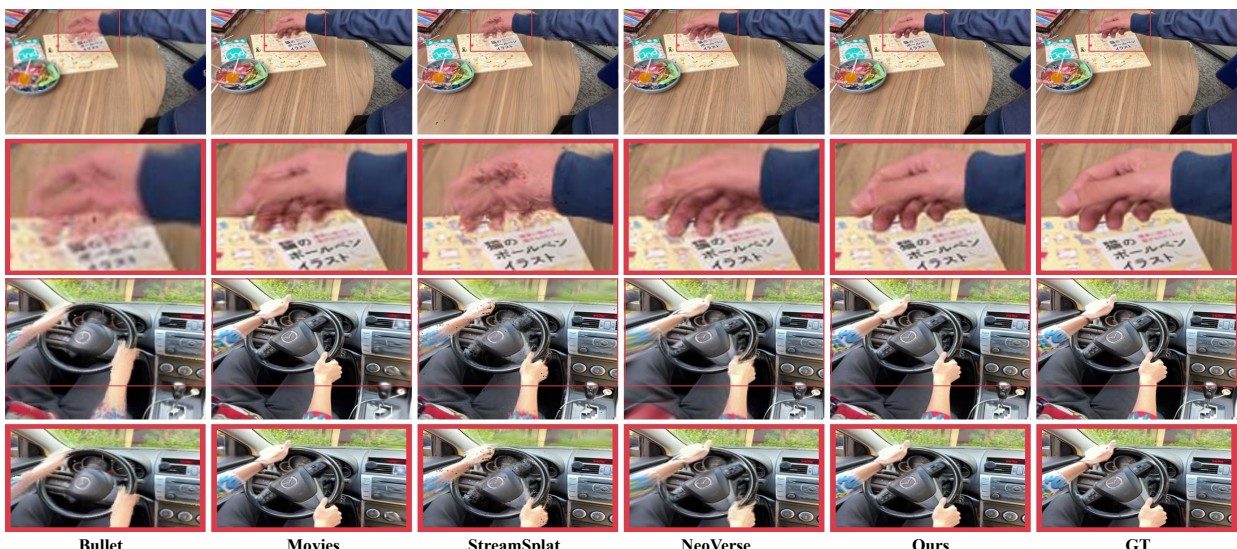

| Bullet | Movies | StreamSplat | NeoVerse | Ours | GT |

*Figure 6.* **Qualitative results on non-uniform motion scenarios.**

*Table 3.* Quantitative results (16-frame intervals) on the SNU-Extreme benchmark for extremely dynamic scenarios (Wu et al., 2025).

| Method | Type | SNU-Extreme 16 Frames | | |
|---|---|---|---|---|
| | | PSNR↑ | SSIM↑ | LPIPS↓ |
| LDMVFI (Danier et al., 2024) | | 20.08 | 0.590 | 0.294 |
| FILM (Reda et al., 2022) | pixel. | 23.71 | 0.728 | 0.171 |
| AMT (Li et al., 2023) | | 24.43 | 0.850 | 0.107 |
| StreamSplat (Wu et al., 2025) | | 19.83 | 0.592 | 0.263 |
| Movies (Lin et al., 2025) | 3DGS. | 21.00 | 0.622 | 0.240 |
| Bullet (Liang et al., 2025a) | | 19.04 | 0.570 | 0.295 |
| NeoVerse (Yang et al., 2026) | | 20.06 | 0.604 | 0.259 |
| Ours | | 23.19 | 0.697 | 0.177 |

*Table 4.* **Ablation study on Gaussian heads with the same backbone.** We evaluate TGs-head, 3DGS+Deform (Wu et al., 2025) and 3DGS+Speed (Yang et al., 2026).

| Setting | GS-Head | | | SNU-HARD 8 frames | | |
|---|---|---|---|---|---|---|
| | 3DGS+Deform | 3DGS+Speed | TGs | PSNR | SSIM | LPIPS |
| (a) | ✓ | | | 21.89 | 0.621 | 0.226 |
| (b) | | ✓ | | 22.00 | 0.639 | 0.220 |
| (c) | | | ✓ | 25.19 | 0.753 | 0.150 |

**Visualization Analysis.** We present several non-uniform motion cases in Fig. 6 to qualitatively evaluate the performance of our method. As illustrated, our model produces high-fidelity renderings with clear boundaries, effectively capturing various objects even under abrupt motion changes. This performance is largely attributed to our supervision strategy. As shown in Fig. 5, directly supervising 3D scene flow often suffers from the inherent coupling of depth and motion (Liang et al., 2025b). In contrast, our decoupled supervision via TGs-Flow provides more accurate motion cues, resulting in significantly fewer artifacts in the synthesized views.

### 5.3. Ablation Studies

In this section, we conduct ablation experiments to evaluate our key design choices: (i) the effectiveness of the TGs

et al., 2025), our model exhibits strong performance in handling long-term non-uniform motion scenarios.

**Acceleration Analysis.** To quantify the impact of non-uniform motion, we analyze the relationship between scene acceleration and PSNR, as shown in Fig. 7. The results illustrate the PSNR performance of various methods under different acceleration scales. The relative stability of our method compared to others provides evidence that accounting for higher-order dynamics effectively mitigates estimation drift in high-acceleration scenarios.

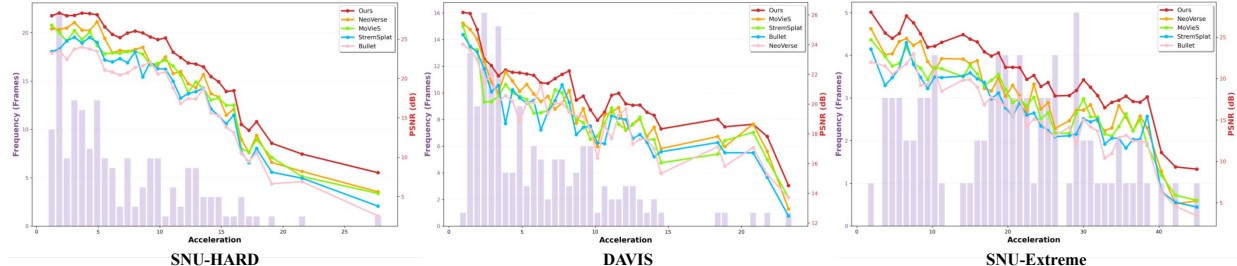

*Figure 7.* **PSNR vs Acceleration.** Higher accelerations correspond to larger non-uniform motion.

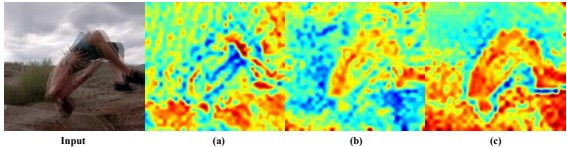

*Figure 8.* **The feature map of Gaussian heads.**

*Table 5.* **Ablation on Expansion Orders for TGs-Flow.**

| | The expansion orders | | | SNU-HARD 8 frames | | |
|---|---|---|---|---|---|---|
| Setting | Zero-order | First-order | Second-order | PSNR | SSIM | LPIPS |
| (a) | ✓ | | | 19.71 | 0.5776 | 0.415 |
| (b) | | ✓ | | 23.28 | 0.698 | 0.188 |
| (c) | | | ✓ | 25.19 | 0.753 | 0.150 |

representation, (ii) the impact of expansion orders, (iii) the influence of optical flow data scaling, and (iv) the contribution of individual loss terms.

**Effectiveness of Taylor-Gaussians Head.** To verify that the TGs-Head effectively captures higher-order motion, we compare it with various baseline heads using the same PAGE backbone (Zhou et al., 2025) (Table 4). As visualized in the feature heatmaps (Fig. 8), the velocity-based baseline (Yang et al., 2026) often fails to represent regions with significant acceleration, whereas our TGs-Head accurately localizes these dynamic areas.

**Impact of Expansion Orders.** We explore the interaction between Taylor expansion orders and flow supervision in Fig. 9. Both quantitative (Table 5) and qualitative results confirm that incorporating the second-order (acceleration) term is essential for capturing non-uniform motion, addressing the limitations of first-order approximations. However, extracting reliable third-order supervision signals from current optical flow models is highly sensitive to temporal noise, leading to unstable training.

**Impact of Indirect Data Scaling.** To investigate how the scale of 2D prior data influences 3D motion synthesis, we conduct an ablation study by employing flow estimators pre-trained on datasets of varying sizes. As shown in Table 6, we observe that enhanced 2D motion priors lead to consistent gains in 3D rendering quality. This result highlights the primary advantage of our decoupled supervision strategy: It allows the framework to indirectly benefit from the data

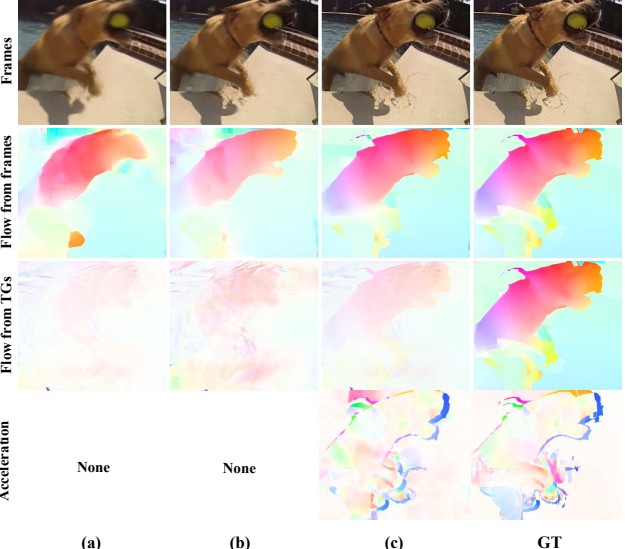

*Figure 9.* **Taylor Expansion Orders for TGs-Flow.**

*Table 6.* **Impact of Indirect Data Scaling**. Quantitative results demonstrate that larger datasets for the flow estimator (Morimitsu et al., 2025) consistently improve motion modeling performance.

| | Optical Flow Dataset | | | | SNU-HARD 8 frames | |
|---|---|---|---|---|---|---|
| Setting | Chairs | Spring | Things | KITTI | PSNR | SSIM |
| (a) | ✓ | | | | 23.61 | 0.663 |
| (b) | ✓ | ✓ | | | 25.19 | 0.753 |
| (c) | ✓ | ✓ | ✓ | | 27.03 | 0.815 |
| (d) | ✓ | ✓ | ✓ | ✓ | 28.63 | 0.857 |

scaling of 2D optical flow, thereby circumventing the inherent scarcity of 3D scene flow labels. Furthermore, while direct scene flow estimation often suffers from error propagation between depth and motion (Liang et al., 2025b), our approach decouples these components, further validating the robustness and practicality of supervising 3D dynamics with large-scale 2D priors, as shown in Table 6 and Fig. 5.

**Loss Function and Regularization.** We evaluate the contribution of each loss term in Table 7. The results show that model performance significantly degrades when the acceleration loss is removed, demonstrating that higher-order temporal constraints are essential to complement first-order flow supervision. The joint optimization of velocity and

*Table 7.* Ablation study on loss function.

| Setting | Loss Function | | | | SNU-HARD 8 frames | | |
|---------|---------------|---|---|---|-------------------|---|---|
| | $\mathcal{L}_{rgb}$ | $\mathcal{L}_f$ | $\mathcal{L}_a$ | $\mathcal{L}_{ref}$ | PSNR | SSIM | LPIPS |
| (a) | ✓ | | | | 20.33 | 0.608 | 0.295 |
| (b) | ✓ | ✓ | | | 23.70 | 0.679 | 0.198 |
| (c) | ✓ | ✓ | ✓ | | 24.86 | 0.738 | 0.163 |
| (d) | ✓ | ✓ | ✓ | ✓ | 25.19 | 0.753 | 0.150 |

*Table 8.* Hyperparameter of $\lambda_f$ and $\lambda_a$ with fixed $\lambda_{reg}$.

| $\lambda_f \setminus \lambda_a$ | 0.50 | 0.60 | 0.70 | 0.80 | 0.90 |
|------|------|------|------|------|------|
| 0.10 | 24.71 | 24.88 | 24.95 | 24.87 | 24.69 |
| 0.20 | 24.85 | 24.90 | **25.10** | 25.07 | 24.82 |
| 0.30 | 24.86 | 24.92 | **25.10** | 25.09 | 24.84 |
| 0.40 | 24.78 | 24.81 | 24.96 | 24.97 | 24.77 |

acceleration losses enables the framework to accurately capture complex non-uniform trajectories.

Furthermore, we validate the necessity of the isotropic regularization term. In dynamic scenarios, certain Gaussians often undergo extreme stretching, leading to geometric degeneration (Yugay et al., 2023). By introducing this regularization, we maintain stable Gaussian shapes during optimization, ensuring the validity of our motion simplification $(B_{i,t}B_{i,t_k}^{-1} \approx I)$ as discussed above. **Hyperparameter analysis.** We conduct a hyperparameter search for the TG-Flow loss terms ($\lambda_f$, $\lambda_a$, and $\lambda_{reg}$). Table 8 and 9 show that fine-tuning these weights yields little promotion, illustrating that our performance improvements do not benefit from hyperparameter tuning.

**Pseudo-supervision quality of TGs-Flow.** We evaluate TGs-Flow on the Sintel (clean) and (final) test sets to assess its robustness. The Sintel (final) set specifically includes complex dynamics like noise and occlusions. Table 10 shows that the model maintains a stable End-Point Error (EPE ↓). The EPE measures the mean Euclidean distance between two optical flows.

## 6. Conclusion

We present **Taylor-Gaussian-Flow (TGsF)**, a novel algorithm designed to address the challenges of long-term non-uniform motion in novel view synthesis. Our approach consists of two core components: **Taylor-Gaussians (TGs)** for explicitly representing high-order motion dynamics, and **TGs-Flow**, which decouples depth and motion to enable effective supervision using widely available 2D optical flow data. Extensive experiments across multiple dynamic benchmarks demonstrate that our method achieves state-of-the-art performance in terms of both visual quality and motion accuracy. Furthermore, detailed ablation studies validate the efficacy of each component within our theoretically grounded framework.

*Table 9.* Hyperparameter analysis of $\lambda_{reg}$ with fixed $\lambda_f$ and $\lambda_a$.

| $\lambda_{reg}$ | 0.00 | 0.05 | 0.10 | 0.15 | 0.20 |
|------|------|------|------|------|------|
| | 25.13 | 25.10 | 25.10 | 25.10 | 25.06 |

*Table 10.* Supervision quality of TGs-Flow.

| | Sintel (clean) test set | Sintel (final) test set |
|---|---|---|
| w First-order | 1.31 | 2.57 |
| w First-order + Second-order | 1.15 | 2.04 |

## Impact Statement

This work advances dynamic novel view synthesis from monocular videos by improving the modeling of long-term non-uniform motion, which may benefit VR/AR, sports broadcasting, robotics, and creative video production. Potential negative impacts include misuse for realistic synthetic media or privacy-intrusive reconstruction, so deployment should consider consent-aware data use, content provenance, and safeguards against deceptive applications.

## Acknowledgment

This work was supported by the National Key R&D Program of China (2022ZD0161800), the National Natural Science Foundation of China under Grant 62271203, AI-Empowered Research Paradigm Reform and Discipline Leap Plan under Grant 2024AI01012 and the Open Research Fund of KLATASDS-MOE, ECNU.

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
