# OpenReview forum: "Taylor-Gaussians-Flow: Towards Non-uniform Motion for Novel View Synthesis from Monocular Video"
_ICML.cc/2026/Conference — ICML 2026 regular_

### Official Review · Reviewer_JasZ · 2026-03-11

**Soundness:** 2
**Presentation:** 3
**Significance:** 3
**Originality:** 3
**Overall Recommendation:** 4
**Confidence:** 3

**Summary:**

The paper addresses the limitations of current feed-forward Novel View Synthesis (NVS) methods in handling long-term, non-uniform dynamic scenes. Existing approaches typically rely on first-order motion approximations (e.g., standard scene flow or linear deformation fields), which fail to capture higher-order kinematics like acceleration. To solve this, the authors propose Feed-Forward Taylor-Gaussians-Flow (FF-TGsF). The core representation, Taylor-Gaussians (TGs), extends 3D Gaussian attributes by introducing a quadratic temporal term for positions and a time-dependent representation for opacity. To train this without requiring scarce ground-truth 3D scene flow data, the authors introduce TGs-Flow, a module that decouples scene-flow supervision into 2D optical flow and depth constraints. The authors report that FF-TGsF achieves state-of-the-art results across four dynamic view-synthesis benchmarks.

**Compliance With Llm Reviewing Policy:**

Affirmed.

**Final Justification:**

The authors have addressed my concerns. I will keep my inital rating.

**Key Questions For Authors:**

see weakness

**Limitations:**

The authors must explicitly discuss the limitations regarding the sensitivity of the loss hyperparameters ($\lambda$). Furthermore, they should acknowledge the potential failure modes of the finite-difference acceleration supervision when dealing with low-quality pre-trained flow estimates and the boundaries of the isotropic assumption.

**Strengths And Weaknesses:**

### Strengths:
1. The fundamental motivation that first-order motion approximations fail under long-term, complex dynamic scenarios is physically sound. Using a Taylor series expansion to naturally introduce a quadratic (acceleration) term to 3D Gaussian trajectories is a logical and mathematically grounded extension.
2. Decoupling scene flow into 2D optical flow and depth is a pragmatic design choice that elegantly sidesteps the lack of robust 3D flow datasets.

### Weaknesses:
1. The reliance on finite-difference approximations for acceleration supervision (Eq. 12) is problematic. Finite difference is notoriously sensitive to noise; since the input is pre-trained optical flow ($\hat{f}$), any temporal jitter or estimation error in the flow model will be amplified by the division by $\Delta t^2$ (as implied by the derivation). The paper lacks an analysis of how this noise impacts the convergence or stability of the acceleration parameters.

2. The method depends on external pre-trained models for pseudo-ground-truth generation (both for flow and acceleration). If the pre-trained flow estimator fails due to occlusions or textureless regions, the TGs-Flow supervision provides garbage gradients, potentially poisoning the training of the TGs Head.

3. The paper lacks ablation studies for the loss weights ($\lambda_f, \lambda_a, \lambda_{reg}$) in Eq. 11. Given the complexity of the multi-task objective, the sensitivity of the model to these hyperparameters is unknown. Without an ablation study, it is impossible to determine if the performance gains are intrinsic to the TGs-Flow method or merely a result of fine-tuning these weights.

---

> ### Author Rebuttal · Authors · 2026-03-30
>
> ## [w1] Noise in finite-difference acceleration supervision.
> We evaluate the acceleration stability on the Sintel (clean) and (final) datasets.
> The (final) set specifically introduces noise and occlusions.
> As the table shows, the model maintains a stable End-Point Error (EPE $\downarrow$) under these noisy conditions.
>
> This confirms that the finite-difference approximation error stays within a manageable range and does not disrupt training convergence.
> The anonymous project page (https://anonymous.4open.science/w/261-re-4444/) provides visual comparisons.
>
> **Sintel test set performance measured by EPE($\downarrow$)**
> | | Sintel (clean) test set | Sintel (final) test set |
> | :---            | :---: | :---: |
> | Ours 1st-order    | 1.31  | 2.57 |
> | Ours 1st-order + 2nd-order| 1.15  | 2.04 |
>
> ## [w2] Robustness to complex motion.
> Our model learns temporal dynamics from the statistical patterns of large-scale data, rather than overfitting to single-frame gradient noise.
> The main text's ablation study already discusses the impact of complex motions, such as occlusions.
> Furthermore, to demonstrate that this gradient noise stays within a manageable range, we evaluate the method zero-shot on diverse motion patterns.
> The anonymous project page (https://anonymous.4open.science/w/261-re-4444/) provides visual comparisons.
>
> The table below shows stable performance across unseen scenarios. This confirms the noise remains well-contained and does not poison the TGs-head training.
>
> **Zero-shot on Four Motion Patterns**
> |   | Abrupt Motion [1]| Long Sequence [2] | View Changes [3]| Discontinuous Motion [4]|
> | :--- | :------------:| :----------:  |:----------:  | :-------------: |
> | Ours  (PSNR $\uparrow$)   | 29.613  | 28.737|21.005 | 26.710 |
> | Ours (LPIPS $\downarrow$) | 0.066   | 0.093 |0.280  | 0.113  |
>
> [1] AMTSet: a benchmark for abrupt motion.
>
> [2] Deep Joint Deblurring and Multi-frame Interpolation with Flow-Guided Attentive Correlation and Recursive Boosting.
>
> [3] Large-scale data for multiple-view stereopsis.
>
> [4] Exploring Discontinuity for Video Frame Interpolation.
>
> ## [w3] Hyperparameter analysis.
> Due to time limitations, we conduct a brief hyperparameter search for the TG-Flow loss terms ($\lambda_f$, $\lambda_a$, and $\lambda_{reg}$).
> The results show that fine-tuning these weights yields only a little promotion
> , illustrating that our performance improvements do not benefit from hyperparameter tuning.
>
> **Hyperparameter analysis of $\lambda_f$ and $\lambda_a$ with fixed $\lambda_{reg}$**
> | $\lambda_a \setminus \lambda_f$ | 0.50 | 0.60 | 0.70 | 0.80 | 0.90 |
> | :------- |  :---- | :---- | :-------- | :---- | :---- |
> | **0.10** |  24.71 | 24.88 | 24.95     | 24.87 | 24.69 |
> | **0.20** |  24.85 | 24.90 | **25.10** | 25.07 | 24.82 |
> | **0.30** |  24.86 | 24.92 | **25.10** | 25.09 | 24.84 |
>
> ***
>
> **Hyperparameter analysis of $\lambda_{reg}$ with fixed $\lambda_f$ and $\lambda_a$**
> | $\lambda_{reg}$ | 0.05  | 0.10 | 0.15  | 0.20 |
> | :-------       | :---- | :---    | :---- | :---- |
> |                 | 25.10 | 25.10 | 25.10 | 25.06 |
>
>
> Furthermore, to clarify whether the performance is obtained from the backbones, we experimented on the SNU-HARD dataset.
> The following table demonstrates that the improvements are primarily attributed to the TG-Flow module.
>
> **Ablation for different backbones on SNU-HARD 8 frames**
> | Setting $\setminus$ Backbones |  VGGT  | CuT3R  | PAGE |
> | :-------    |  :---: | :---: | :---:|
> |   w TG-Flow |  24.98 | 24.90 | 25.10|
> | w/o TG-Flow |  20.03 | 20.19 | 19.71|
>
>
> In addition, Table 4 and Figure 8 in the main text compare different Gaussian heads on the same baseline.
> These results show the gains do not come from stronger baselines.

---

> > ### Author Rebuttal · Reviewer_JasZ · 2026-04-05
> >
> > The authors did an excellent job by addressing all my three concerns

---

### Official Review · Reviewer_2izG · 2026-03-11

**Soundness:** 3
**Presentation:** 3
**Significance:** 3
**Originality:** 3
**Overall Recommendation:** 4
**Confidence:** 4

**Summary:**

Summary

This paper proposes Feed-Forward Taylor-Gaussians-Flow (FF-TGsF) for novel view synthesis from monocular video, with a focus on addressing the challenges of modeling non-uniform motion. The core idea is to formulate the motion of dynamic Gaussians using a Taylor expansion, explicitly modeling velocity and acceleration in the representation, while incorporating temporal opacity to handle time-varying visibility.

In addition, the paper introduces TGs-Flow as a supervision signal, arguing that compared with directly supervising 3D scene flow, the decoupled supervision is more stable and leads to fewer artifacts.

Based on the experimental description, the authors fine-tune the model on DAVIS and RE10K, treating them as video datasets that do not rely on pre-calibrated camera information.

**Compliance With Llm Reviewing Policy:**

Affirmed.

**Key Questions For Authors:**

How robust is the method to optical-flow errors in the pseudo-labels?

**Strengths And Weaknesses:**

Strengths

1.The problem is well-motivated and meaningful.The paper focuses on the challenging issue of non-uniform motion in monocular video, which is clearly defined and practically important.

2. The method design is natural. Explicitly modeling velocity and acceleration through a Taylor expansion is an intuitive and reasonable extension of dynamic 3DGS. The introduction of temporal opacity also fits well with the needs of dynamic scene modeling.

3. The supervision design is convincing. The paper points out that directly supervising 3D scene flow can be affected by the coupling between depth and motion, and TGs-Flow is designed to alleviate this issue. This motivation is reasonable.

4. The ablation results are encouraging. According to the ablation studies, the TGs-head shows clear improvements over several baselines, suggesting that the key design choices do contribute meaningful performance gains.

Weaknesses

1. The analysis of higher-order motion modeling could be more fine-grained. While the paper already provides ablations on the expansion order, a more detailed breakdown across different motion patterns would further clarify when the second-order term is most beneficial.

2. The robustness to pseudo-supervision quality could be discussed more explicitly. The supervision design is well motivated, but it still relies on estimated optical flow as pseudo supervision. Additional discussion on robustness under noisy flow estimation would make the method more convincing.

---

> ### Author Rebuttal · Authors · 2026-03-30
>
> Thank you for your insightful comments, and we will provide detailed responses to each of them below:
>
> ## [W1] Fine-grained analysis of the second-order term under different motion patterns.
> We evaluate our model on abrupt motion, long sequences, view changes, and discontinuous motion.
> The anonymous project page (https://anonymous.4open.science/w/261-re-4444/) provides the corresponding visualizations.
>
> Results show the second-order (acceleration) term significantly improves view synthesis for abrupt motions and long sequences.
> However, it yields no gains for discontinuous motion, as acceleration inherently assumes continuous physical dynamics.
>
>
> **Abrupt Motion [1]**
> | Method | Ours 1-order | Ours 1-order + 2-order | StreamSplat | MoVieS |
> |--------|-----------|--------|----------------|------------------|
> | PSNR($\uparrow$)  | 27.046 | **29.613** | 25.020 | 25.209 |
> | LPIPS($\downarrow$) |0.106 | 0.066 | 0.155 | 0.149 |
>
> ***
>
> **Long Sequences of YouTube [2]**
> | Method | Ours 1-order | Ours 1-order + 2-order| StreamSplat | MoVieS |
> |--------|-----------|--------|----------------|------------------|
> | PSNR($\uparrow$)   | 26.861 | **28.737** | 24.704 | 24.011 |
> | LPIPS($\downarrow$) |0.110 | 0.093 | 0.164 | 0.187 |
>
> ***
>
> **View Changes [3]**
> | Method | Ours 1-order  | Ours 1-order + 2-order | StreamSplat | MoVieS |
> |--------|-----------|--------|----------------|------------------|
> | PSNR($\uparrow$)  | 20.567 | **21.005** | 19.502 | 19.960 |
> | LPIPS($\downarrow$)| 0.388 | 0.280 | 0.431 | 0.396 |
>
> ***
>
> **Discontinuous Motion [4]**
> | Method | Ours 1-order  | Ours 1-order + 2-order | StreamSplat | MoVieS |
> |--------|-----------|--------|----------------|------------------|
> | PSNR($\uparrow$)  | 26.501 | 26.61 | 26.283 | 26.315 |
> | LPIPS($\downarrow$)|0.117 | 0.113 | 0.122 | 0.121 |
>
> [1] AMTSet: a benchmark for abrupt motion.
>
> [2] Deep Joint Deblurring and Multi-frame Interpolation with Flow-Guided Attentive Correlation and Recursive Boosting.
>
> [3] Large-scale data for multiple-view stereopsis.
>
> [4] Exploring Discontinuity for Video Frame Interpolation.
>
>
> ## [w2 & Q1] Robustness to optical flow pseudo-supervision quality.
> We evaluate TGs-Flow on the Sintel (clean) and (final) test sets to assess its sensitivity.
> The Sintel (final) set specifically includes complex dynamics like noise and occlusions.
> As the table shows, the model maintains a stable End-Point Error (EPE $\downarrow$).
> The EPE measures the mean Euclidean distance between two optical flows.
>
> Both the 1st-order and 2nd-order formulations demonstrate strong robustness under challenging motions.
> The anonymous page (https://anonymous.4open.science/w/261-re-4444/) provides GIF visualizations.
>
> **Sintel test set performance measured by EPE($\downarrow$)**
> | | Sintel (clean) test set | Sintel (final) test set |
> | :---            | :---: | :---: |
> | Ours 1st-order    | 1.31  | 2.57 |
> | Ours 1st-order + 2nd-order| 1.15  | 2.04 |
> | MoVieS (3D motion field) | 1.93  | 3.54 |

---

> > ### Author Rebuttal · Reviewer_2izG · 2026-04-03
> >
> > Thank you! The review have resolved all my concerns.

---

### Official Review · Reviewer_j9jC · 2026-03-12

**Soundness:** 3
**Presentation:** 3
**Significance:** 3
**Originality:** 3
**Overall Recommendation:** 4
**Confidence:** 3

**Summary:**

The paper proposes Taylor-Gaussians-Flow, a feed-forward framework designed to model non-uniform motion for novel view synthesis from monocular videos. The method leverages a Taylor-based parameterization of Gaussian motion and predicts scene representation in a feed-forward manner instead of relying on iterative optimization. The goal is to improve efficiency while handling complex motion patterns. Experimental results show that the approach achieves competitive rendering quality and improved inference efficiency compared with optimization-based Gaussian approaches.

**Compliance With Llm Reviewing Policy:**

Affirmed.

**Key Questions For Authors:**

- How sensitive is the model to the order of the Taylor expansion?
- How well does the method generalize to scenes with very abrupt motion or motion discontinuities?
- Can the proposed framework scale to longer monocular videos with larger viewpoint changes?

**Limitations:**

- The method may rely on assumptions about motion smoothness due to the Taylor approximation.
- Performance in highly dynamic scenes or scenes with strong occlusion changes is not fully analyzed.

**Strengths And Weaknesses:**

## Strengths
- The paper addresses an important problem in dynamic scene reconstruction and novel view synthesis, particularly the challenge of modeling non-uniform motion.
- The proposed Taylor-based formulation for Gaussian motion is conceptually interesting and provides a structured way to approximate motion dynamics.
- The feed-forward design improves inference efficiency compared with optimization-based 3D Gaussian pipelines.
- The experimental results demonstrate promising performance on several datasets.

## Weaknesses
- The technical description of the Taylor-based motion formulation could be clearer, particularly regarding the assumptions behind the approximation.
- The method’s relationship to recent feed-forward 3D Gaussian approaches is not fully discussed.
- Some experimental comparisons are limited, and stronger baselines could better contextualize the improvements.
- The ablation analysis on motion modeling choices and Taylor order is relatively limited.

---

> ### Author Rebuttal · Authors · 2026-03-30
>
> Thank you for your insightful suggestions, and we will provide detailed responses to each of them below:
>
> ## [W1] Clarity of the Taylor-based formulation.
> To improve clarity, we will complete the method section and move the detailed formulation of Taylor-GS from the supplementary material directly into the main text.
>
> ## [w2 ] Relationship to feed-forward 3D Gaussian approaches.
> We will discuss this relationship in the experimental section of the main text.
> In addition, experimental analyses and discussions about different motion patterns will also be included, as presented in section [W4].
>
> ## [w3] Is the performance improvement caused by the baseline?
>  Table 4 and Figure 8 in the main text compare different Gaussian heads on the same baseline. These results show the gains do not come from stronger baselines.
>
> Furthermore, to clarify whether the performance is obtained from the backbones, we experimented on the SNU-HARD dataset.
> The table shows that the improvements are primarily attributed to the TG-Flow module.
>
> **Ablation for different backbones on SNU-HARD 8 frames**
> | Setting $\setminus$ Backbones |  VGGT  | CuT3R  | PAGE |
> | :-------    |  :---: | :---: | :---:|
> |   w TG-Flow |  24.98 | 24.90 | 25.10|
> | w/o TG-Flow |  20.03 | 20.19 | 19.71|
>
> ## [W4 & Q2 & Q3] Robustness under different motion patterns.
> We evaluate our model on abrupt motion, long sequences, view changes, and discontinuous motion.
> The anonymous project page (https://anonymous.4open.science/w/261-re-4444/) provides the corresponding visualizations.
>
> The results show our model performs robustly across these datasets.
> Specifically, the acceleration term significantly improves performance on abrupt motions and long sequences.
> However, it brings no gains to discontinuous motion, as acceleration inherently assumes continuous physical movement.
>
> **Abrupt Motion [1]**
> | Method | Ours 1-order | Ours 1-order + 2-order | StreamSplat | MoVieS |
> |--------|-----------|--------|----------------|------------------|
> | PSNR($\uparrow$)  | 27.046 | **29.613** | 25.020 | 25.209 |
> | LPIPS($\downarrow$) |0.106 | 0.066 | 0.155 | 0.149 |
>
> ***
>
> **Long Sequences of YouTube [2]**
> | Method | Ours 1-order | Ours 1-order + 2-order| StreamSplat | MoVieS |
> |--------|-----------|--------|----------------|------------------|
> | PSNR($\uparrow$)   | 26.861 | **28.737** | 24.704 | 24.011 |
> | LPIPS($\downarrow$) |0.110 | 0.093 | 0.164 | 0.187 |
>
> ***
>
> **View Changes [3]**
> | Method | Ours 1-order  | Ours 1-order + 2-order | StreamSplat | MoVieS |
> |--------|-----------|--------|----------------|------------------|
> | PSNR($\uparrow$)  | 20.567 | **21.005** | 19.502 | 19.960 |
> | LPIPS($\downarrow$)| 0.388 | 0.280 | 0.431 | 0.396 |
>
> ***
>
> **Discontinuous Motion [4]**
> | Method | Ours 1-order  | Ours 1-order + 2-order | StreamSplat | MoVieS |
> |--------|-----------|--------|----------------|------------------|
> | PSNR($\uparrow$)  | 26.501 | 26.61 | 26.283 | 26.315 |
> | LPIPS($\downarrow$)|0.117 | 0.113 | 0.122 | 0.121 |
>
> [1] AMTSet: a benchmark for abrupt motion.
>
> [2] Deep Joint Deblurring and Multi-frame Interpolation with Flow-Guided Attentive Correlation and Recursive Boosting.
>
> [3] Large-scale data for multiple-view stereopsis.
>
> [4] Exploring Discontinuity for Video Frame Interpolation.
>
>
> ## [Q1]  How sensitive is the model to the order of the Taylor expansion?
> We evaluate TGs-Flow on the Sintel (clean) and (final) test sets to assess its sensitivity.
> The Sintel (final) set specifically includes complex dynamics like noise and occlusions.
> As the table shows, the model maintains a stable End-Point Error (EPE $\downarrow$).
> The EPE measures the mean Euclidean distance between two optical flows.
>
> Both the 1st-order and 2nd-order formulations demonstrate strong robustness under challenging motions. The anonymous page (https://anonymous.4open.science/w/261-re-4444/) provides GIF visualizations.
>
> **Sintel test set performance measured by EPE($\downarrow$)**
> | | Sintel (clean) test set | Sintel (final) test set |
> | :---            | :---: | :---: |
> | Ours 1st-order   | 1.31  | 2.57 |
> | Ours 1st-order + 2nd-order| 1.15  | 2.04 |

---

> > ### Author Rebuttal · Reviewer_j9jC · 2026-04-07
> >
> > Thanks for the authors' response. I will keep my rating.

---

### Official Review · Reviewer_ZX99 · 2026-03-12

**Soundness:** 2
**Presentation:** 2
**Significance:** 3
**Originality:** 2
**Overall Recommendation:** 2
**Confidence:** 4

**Summary:**

This paper aims to solve feed-forward monocular video reconstruction problem, achieving good novel view synthesis results. The main contribution is their Taylor-Gaussians representation and optical-flow based training scheme. They achieve promising results on all the proposed datasets.

**Compliance With Llm Reviewing Policy:**

Affirmed.

**Final Justification:**

I have carefully considered the authors' rebuttal and further responses and re-evaluated the manuscript. My final decision is outlined below.

The core issue lies in the fundamental dependency of the proposed method on the pre-trained "PAGE-4D." The authors perform fine-tuning exclusively on the middle layers related to mask-aware attention. Consequently, the entire 3D geometric reconstruction capability of their model is inherently derived from PAGE-4D. Based on the authors' response, two critical scenarios emerge:
1. If the fine-tuning process alters PAGE-4D's feature distribution for geometric modeling, it breaks the model's inherent ability to unify point clouds across frames into a consistent coordinate system, leading to potential instability.
2. If the fine-tuning does not alter PAGE-4D's geometric understanding, then the reconstruction performance is solely credited to PAGE-4D, rendering the authors' proposed methodology an add-on.

Thus, the novelty of this work is relatively narrow. It essentially consists of only two points: (1) the application of flow loss for motion supervision, and (2) a specific fine-tuning trick to adapt a 4D point cloud foundation model into 4D Gaussian model, enabling novel-view synthesis. This positions the research as an incremental extension rather than a fundamental methodological breakthrough.

A major flaw is the complete lack of generalization analysis. The designed fine-tuning strategy is tailor-made for PAGE-4D's specific dynamic mask attention structure. The authors provide no experimental evidence whatsoever to validate whether this trick works on other distinct foundation models. This absence of cross-model testing severely undermines the broader impact and applicability of their claims.

Finally, the authors attempt to validate their approach via an ablation study, where they show poor performance of PAGE-4D without their velocity/acceleration terms. However, this experiment fails to address the key concern raised by reviewer **j9jc**: since the reconstruction ability fundamentally comes from PAGE-4D, the low baseline score of the original foundation model does not prove that their fine-tuning trick is the primary contributor to the final high performance.

In conclusion, the manuscript of this version doesn't really show strong academical contribution, so I am sorry I cannot make a positive recommendation at the current stage.

**Key Questions For Authors:**

Please refer to the weaknesses. Furthermore, I have additional questions as follows:
1. How do you split DAVIS dataset? This dataset contains 90 videos in total, but you mention it contains 30 videos only in [Line324 right].
2. How do you get the camera poses of the DAVIS Middle-8 Frames and SNU-HARD Middle-8 Frames?

As stated in weaknesses, I cannot recommend this paper currently. However, if the authors can address my concerns, I am likely to modify my scores significantly.

**Limitations:**

No limitation discussed. The authors should discuss the sensitivity of the complexity of the motion and the accuracy of their learned flows.

**Strengths And Weaknesses:**

Strengths:
1) the task is very important for the community
2) The method shows great performance, especially exhibit high-fidelity results
3) Solid ablation studies

Weaknesses:
Depite the strengthes, there are also some weaknesses:
1) In your whole pipeline, there is no 3D supervision or camera estimation as output or losses, which means there is no guarantee that the output is in a unified 3D space. More possibly, the depth points estimation part of the methods could degenerate to a monocular depth estimation head. If so, the whole pipeline becomes a simple volume video representation, or in other word a monocular depth estimation with appearance, but no 3D coordinates guaranteed.
2) The training scheme is not well presented. Following weakness 1), your supervision relies on splatted results, but not 3D correspondance, which means you need a camera pose to do the splatting. But in your experiment, you mentioned you don’t need a calibrated camera for supervision, but there is no camera predicted either, so how do you train your model? I can only guess you simply use an idensity camera poses with fixed intrinsics, and the pose is absorbed in the velocity and acceleration. If so, how you align your predicted gaussians into the global scale of the evaluation dataset? Do you use the GT camera poses to align it?
3) There is no quantitative results of the accuracy between your flow predicted by your velocity and acceleration and the flow estimated by the teacher model. This is a direct measure about the accuracy of your learned motion.

In conslusion, I appreaciate the great performance. However, based on the above unclear points and weaknesses, currently I don’t think this paper can serve as a 4D reconstruction pipeline but degenerate to a monocular depth video pipeline. Comparing the visual quality of it with other reconstruction pipeline is unfair, since they reconstruct things in a unified 3D coordinates. So I cannot recommend this paper in the current stage.

---

> ### Author Rebuttal · Authors · 2026-03-30
>
> We appreciate your invaluable insights and thoughtful comments.
> In the following sections, we pleasure to answer the questions you have raised:
>
> # [W1] Camera pose.
> Our model follows the **implicit pose estimation** idea of StreamSplat.
> The camera pose is effectively absorbed into the motion fields.
> Specifically, the model predicts a motion field that transforms the canonical 3D Gaussians to align with the current observation.
> This allows us to handle complex camera trajectories (and even non-rigid camera distortions like rolling shutter).
> This avoids relying on potentially unstable pose estimation in monocular dynamic videos and makes the method applicable even when calibration is unavailable or unreliable.
>
> Visualizations on the anonymous website (https://anonymous.4open.science/w/261-re-4444/) compare implicit and explicit pose estimation.
> The results show that the implicit approach yields greater stability than explicit methods for objective motion scenes.
>
> All compared methods in the main text specifically target the monocular video task.
> As summarized below, most methods avoid explicit camera pose loss terms.
> Our approach (following StreamSplat) utilizes implicit estimation.
>
> | | StreamSplat | MoVieS | Bullet | NeoVerse |
> | :--- | :---: | :---: | :---: | :---: |
> | Monocular video task | **✓** | **✓** | **✓** | **✓** |
> | Camera pose loss term| **✗** | **✗** | **✗** | **✓** |
>
> # [w2 & Q2] Camera pose in Gaussian Splatting and interpolation time.
> **[Re:]**
> We define the implicit camera pose as a global parameter outside the render function.
> Within the function, we derive the Gaussian parameters for timestamps based on interpolation_time.
> The pseudocode is provided below:
> ```python
> R_fixed = np.array([[ 1., -0., 0.],
>                 [ 0., -0., 1.],
>                 [ 0., -1., 0.]])
> T_fixed = np.array([0., 0., 0.])
>
> def getWorld2View2(R, t, translate=np.array([.0, .0, .0]), scale=1.0):
>     Rt = np.zeros((4, 4))
>     Rt[:3, :3] = R.transpose()
>     Rt[:3, 3] = t
>     Rt[3, 3] = 1.0
>     C2W = np.linalg.inv(Rt)
>     cam_center = C2W[:3, 3]
>     cam_center = (cam_center + translate) * scale
>     C2W[:3, 3] = cam_center
>     Rt = np.linalg.inv(C2W)
>     return Rt
>
> def getOrthProjectionMatrix():
>     znear, zfar = 0., 10.0
>     top = 1
>     bottom = -top
>     right = 1
>     left = -right
>     P = torch.zeros(4, 4)
>     z_sign = 1.0
>     P[0, 0] = 2.0 / (right - left)
>     P[1, 1] = 2.0 / (top - bottom)
>     P[2, 2] = -2.0 * z_sign / (zfar - znear)
>     P[0, 3] = -(right + left) / (right - left)
>     P[1, 3] = -(top + bottom) / (top - bottom)
>     P[2, 3] = -(zfar + znear) / (zfar - znear)
>     P[3, 3] = 1.0
>     return P
>
> world_view_transform = getWorld2View2(R_fixed, T_fixed, np.array([0.0, 0.0, 0.0]), 1.0)   # View matrix
> camera_center = world_view_transform.inverse()[3, :3]                                     # Camera position
> projection_matrix = getOrthProjectionMatrix()
> full_proj_transform = world_view_transform.bmm(projection_matrix)                         # Projection matrix
> tanfovx = math.tan(math.pi / 4.0)
> tanfovy = math.tan(math.pi / 4.0)
>
> def render (Gaussians_parameters, interpolation_time):
>     # Camera_pose_parameters
>     view_matrix = world_view_transform.float()       # View matrix
>     view_proj_matrix = full_proj_transform.float()   # Projection matrix
>     campos = camera_center.float()                   # Camera position
>
>     # interpolation_time \in (0-1), Gaussians_parameters
>     # Temporal opacity.
>     trbfdistance = interpolation_time / torch.exp(0.5*trbfscale)
>     trbfoutput = basicfunction(trbfdistance)
>     opacity = pointopacity * trbfoutput
>     pc.trbfoutput = trbfoutput
>
>     # Taylor motion.
>     tforpoly = interpolation_time.detach()
>     velocity = pc._motion[:, 0:3]
>     acceleration = pc._motion[:, 3:6]
>     means3D = means3D + velocity * tforpoly + 0.5 * acceleration * tforpoly * tforpoly
>     .......
>
>     raster_settings = GaussianRasterizationSettings(Gaussians_parameters, Camera_pose_parameters)
>     ......
> ```
>
>
> ## [w3] Quantitative accuracy of optical flow.
> We evaluate the End-Point Error (EPE $\downarrow$) between the optical flow predicted by our TGs-Flow and the teacher model.
> The EPE measures the mean Euclidean distance between two optical flows.
> The low errors in the table below confirm our model's robustness, which stems from the teacher model's large-scale optical flow datasets.
>
> **TGs-Flow vs Teacher Model**
> |             | DyCheck iPhone | SNU-FILM-HARD | SNU-FILM-Extreme |
> | :---        | :---: | :---:  | :---: |
> | Ours 1st-order VS Teacher 1st-order| 0.181  | 0.312 | 0.470 |
> | Ours 2nd-order VS Teacher 2nd-order| 0.198  | 0.376  | 0.495 |
>
> ## [Q1] Split of the DAVIS dataset.
>  As noted in [Line 324 (right)], this information appears in the "Testing Datasets" section of the main text. ‘30 videos’ refers to the official test split of the DAVIS dataset.

---

> > ### Author Rebuttal · Reviewer_ZX99 · 2026-04-01
> >
> > Thanks for the authors' response. Based on the response, I am now sure I have the right understanding of what the paper is doing. I assumed "the pose is absorbed in the velocity and acceleration" in weakness part and the author accept it by calling it "implicit pose estimation". This is the concensus. And this is where my concern is. Let me explain my concern in more detail:
> > 1. As the model uses a so-called fixed canonical representation, that means the reconstructions at different key frames are not aligned in a 3D space. Thus everything becomes relative. My concern lies in whether this is true 3D (or 4D) reconstruction?
> > 2. Following 1, for a novel-view synthesis application (NVS is the core task for this paper), people always provide a target sequence in the world coordinate, but not a relative sequence for the selected key-frames. That means, after running the model, if I want to render the reconsruction in a target world-coordinated camera trajectory, I need to first estimate the camera pose of the given video, then calculate the relative poses between my target and the input views, and finally render the results. In this way, the 3D recontruction is only partially solved.
> > 3. Following 2, this is why I raise the question 2 in weakness "how you align your predicted gaussians into the global scale of the evaluation dataset? Do you use the GT camera poses to align it?" which is not fully responded due to my unclearness. In order to render your results to a given gt camera in world coordinate in DyCheck iphone dataset, do you align the target poses into your canonical pose with the given synchronized ground-truth iphone poses?
> > 4. In response 1, except StreamSplat, the author argues there is no camera loss in the other three baseline models. However, forcing the reconstruction in a unified world coordinated does not require camera losses actually. In their training, they are splatting or rendering with synchronized world-coordinated cameras, thus their results are forced to be within a unified coordinate.
> >
> > Generally speaking, I am not challenging the performance of the paper. I'm just doubting whether this is true 3D reconstruction and whether the required NVS setting can be satisfied in our daily life applciation. So I will not change my recommendation only if the authors can provide a promising way to solve this synchronizing problem.
> >
> > Furthermore, after reading the response and concerns of other reviewers, I want to further raise a question, but to be fair for the authors, this question (or weakness) will not influence my rating: since the model requires a smooth motion (for acceleration, "acceleration inherently assumes continuous physical movement" in response to j9jC), I wonder the performance of the model for a video with unstable camera trajectories, such as noise due to hand shaking, which is common in daily life.

---

> > > ### Author Response · Authors · 2026-04-02
> > >
> > > # [Q1 & Q2] Regarding True 3D Reconstruction.
> > > **Re:**
> > > We thank the reviewer for these professional comments to improve the manuscript.
> > >
> > > *Summary.*  We fine-tune four dynamic attention layers of PAGE as our backbone.
> > > The point clouds from PAGE are globally aligned and maintain 3D reconstruction capabilities.
> > > On datasets like Dycheck iPhone, we use an explicit camera pose strategy.
> > > For video interpolation datasets, we use an implicit camera pose strategy to avoid potentially unstable pose estimation.
> > > Experiments in the main text show that both explicit and implicit strategies perform well. Finally, the camera strategy is not the primary innovation of this paper.
> > >
> > >
> > > *Fine-tuning Strategy.*
> > > Estimating motion and camera poses in dynamic scenes is an inherent conflict between tasks.
> > > Accurate camera pose estimation requires suppressing dynamic regions, while geometry reconstruction requires modeling them.
> > > In other words, accurate pose estimation requires static regions, while dynamic objects introduce errors.
> > > To this end, we follow the strategy and freeze other parameters of the PAGE model.
> > > We only fine-tune four Mask Attention layers responsible for dynamic regions.
> > > This approach preserves the camera poses and the global spatial alignment of static regions within the 3D foundation model.
> > > Totally, the fine-tuning aims to improve dynamic representation without damaging static alignment or camera pose accuracy.
> > >
> > > *Initialization.*
> > > We use the 3D point cloud coordinates from PAGE to initialize the 3D Gaussian means.
> > > Because PAGE maintains camera pose consistency and static alignment, the resulting 3D Gaussians represent a true 3D reconstruction.
> > >
> > > # [Q3] How you align your predicted gaussians into the global scale of the evaluation dataset? Do you use the GT camera poses to align it?
> > >
> > > **Re:**
> > > We thank the reviewer for patient and detailed explanations.
> > >
> > > Our framework fine-tunes only the four Mask Attention layers (dynamic regions) of the PAGE model.
> > > This process does not disrupt the camera poses and the static region alignment within the original model.
> > > Consequently, the 3D Gaussians based on the PAGE output maintain global scale alignment.
> > > On DyCheck iphone dataset,  we utilize the poses predicted by the PAGE model.
> > >
> > >
> > > # [Q5] Camera shaking.
> > > **Re:** We thank the reviewer for the insightful comments.
> > > We are willing to test the performance in this scene.
> > >
> > > We conduct a simple test on the GoPro dataset to evaluate the scene.
> > > The GoPro dataset consists of videos recorded by pedestrians holding cameras while walking, which contains significant camera shake.
> > > The table below demonstrates that the second-order term provides some improvement, though the gain is less significant than in scenarios with abrupt motion.
> > >
> > > **Gopro PSNR($\uparrow$)**
> > > |  Methods | Ours 1st-order |Ours 1st-order + 2nd-order | StreamSplat| MoVieS|
> > > | :----    | :------------: | :-------------------: | :--------: | :---: |
> > > |          | 23.660         |  24.081                |  23.218   | 22.971|
> > >
> > > **Gopro LPIPS($\downarrow$)**
> > > |  Methods |  Ours 1st-order|Ours 1st-order + 2nd-order | StreamSplat| MoVieS|
> > > | :----    | :------------: | :-------------------: | :--------: | :---: |
> > > |          | 0.199	        |       0.186	        |   0.216	 | 0.226 |

---

### Decision · Program_Chairs · 2026-04-30

**Decision:**

Accept (regular)

**Comment:**

This paper received mixed scores of 1 reject and 3 weak accepts. All the reviewers agree that the task is important and experiments are solid. However, there is an important concern left about the true contribution over the backbone PAGE-4D of this paper. After reading the paper, review comments, rebuttal and also the discussions of the reviewers, the AC recommends acceptance. Since this paper targets at an important task and also the experiments are promising, the AC believes that the community would learn and benefit from this paper. The authors should carefully address the concerns especially about the novelty over PAGE-4D + velocity head.